# Resveratrol and Tumor Microenvironment: Mechanistic Basis and Therapeutic Targets

**DOI:** 10.3390/molecules25184282

**Published:** 2020-09-18

**Authors:** Wamidh H. Talib, Ahmad Riyad Alsayed, Faten Farhan, Lina T. Al Kury

**Affiliations:** 1Department of Clinical Pharmacy and Therapeutic, Applied Science Private University, Amman 11931-166, Jordan; a_alsayed@asu.edu.jo (A.R.A.); ph.fatenfarhan@yahoo.com (F.F.); 2Department of Health Sciences, College of Natural and Health Sciences, Zayed University, Abu Dhabi 144534, UAE; Lina.AlKury@zu.ac.ae

**Keywords:** resveratrol, tumor microenvironment, natural products, anti-cancer, phytotherapy

## Abstract

Resveratrol (3,4′,5 trihydroxystilbene) is a naturally occurring non-flavonoid polyphenol. It has various pharmacological effects including antioxidant, anti-diabetic, anti-inflammatory and anti-cancer. Many studies have given special attention to different aspects of resveratrol anti-cancer properties and proved its high efficiency in targeting multiple cancer hallmarks. Tumor microenvironment has a critical role in cancer development and progression. Tumor cells coordinate with a cast of normal cells to aid the malignant behavior of cancer. Many cancer supporting players were detected in tumor microenvironment. These players include blood and lymphatic vessels, infiltrating immune cells, stromal fibroblasts and the extracellular matrix. Targeting tumor microenvironment components is a promising strategy in cancer therapy. Resveratrol with its diverse biological activities has the capacity to target tumor microenvironment by manipulating the function of many components surrounding cancer cells. This review summarizes the targets of resveratrol in tumor microenvironment and the mechanisms involved in this targeting. Studies discussed in this review will participate in building a solid ground for researchers to have more insight into the mechanism of action of resveratrol in tumor microenvironment.

## 1. Introduction

Cancer is one of the main causes of death globally and expected to be the most significant barrier to increase life expectancy in the 21st century [1]. Cancer is considered as the first or second leading cause of death for people aged below 70 years in 91 countries. Statistics from another 22 countries revealed that cancer is the third or fourth leading cause of mortality [2]. Cancer affects both sexes equally and a total number of 12.7 million cases were estimated in 2015. An increase in this number to 21 million is expected by 2030 [3].

The link between cancer and dietary natural products was documented in many studies. Dietary and lifestyle measures can prevent 30–40% of all cancers [4]. Different regions throughout the world exhibited low incidence of some cancers. The main reasons for such low incidence are dietary habit and the wide use of natural products as a complementary medicine to treat cancer [5,6,7]. These observations encouraged many researchers to study the potential of natural products alone or in combinations to treat different cancers. Natural products like curcumin, piperine, thymoquinone, geneistein and parthenolide are few examples of many phytochemicals tested for their anti-cancer effects [8,9,10,11,12,13]. Resveratrol is a powerful antioxidant presents in high concentrations in grape seeds and skin. It has various biological activities and the research interest in this compound is in continuous rise. Based on PubMed data, the number of published articles focusing on resveratrol is 571 in 2009. This number increased to 1386 in 2019 (approximately 143% increase). The number of articles dealing with resveratrol and cancer is 347 during 2019 only which represent 25% of resveratrol articles published in this year. These numbers showed the importance of resveratrol as a health promoting agent and its essential role in cancer prevention and treatment.

The tumor microenvironment is a complex and continuously changing matrix surrounding cancer cells. It has different kinds of cells that play an important role in cancer progression and development. Fibroblasts, stromal and endothelial cells were detected in tumor microenvironment. In addition to these cells, many innate and acquired immune cells were also observed as a permanent resident in tumor microenvironment [14].

These cells and their products were the target for many anti-cancer therapies. The diverse biological activities of resveratrol make it one of the promising therapies that may regulate tumor microenvironment. In this review, we summarized the targets of resveratrol in tumor microenvironment and the mechanisms by which resveratrol can regulate the functions of these targets.

## 2. Resveratrol: Sources, Bioavailability, Absorption and Metabolism

Resveratrol (*trans*-3,4′,5-trihydroxystilbene) is one of the naturally occurring phytoalexin belonging to the stilbene class. It is synthesized de novo by plants in response to injury or when the plant is under the attack of a pathogen such as bacteria or fungi [15]. Although it is produced naturally by 72 different plants, the primary food sources of resveratrol include wine, grapes, peanut, pomegranate, pines, cocoa, cranberries and dark chocolate [16]. Resveratrol exists in two principal isomers: *Cis* and *trans* (Figure 1) and they both often exist together. However, the trans form is more biologically active [17].

In recent years, several reports have showed that resveratrol is a multipurpose compound. It plays a potentially important role in preventing or treating chronic diseases. Among its effects are antioxidative, anti-inflammatory, anti-proliferative and anti-angiogenesis along with improved cardiovascular outcomes [16,18]. The pharmacokinetics of resveratrol must be determined before any conclusion can be drawn on the health benefits of dietary or commercially available resveratrol. Therefore, it has been the focus of numerous in vivo and in vitro studies to examine its absorption, metabolism and bioavailability. These studies showed that cellular and molecular targets of resveratrol could be changed due to modifications in its structure, stability and metabolism. Such modifications are essential to improve resveratrol chemopreventive effects [19,20].

Intestinal absorption of resveratrol is similar to other flavonoids and isoflavonoid. The aglycon form of resveratrol is taken up by enterocyte through the diffusive process, while the glycosidic form is likely to deglycosylate at the intestinal level [19]. According to urinary excretion data, the absorption extent of resveratrol reach at least 70% [20].

Studies on human and animals were conducted to assess the absorption of resveratrol. In an animal study, the rats were administered with a known resveratrol concentration (6.5 mg/L) in red wine. The study was conducted on 84 male rats and two doses of resveratrol were tested (86 µg/kg and 43 µg/kg). The results of this study demonstrates that resveratrol in wine was quickly absorbed, reaching its peak concentration approximately 60 min after wine ingestion and it was detectable in the liver and kidneys after 30 min [21]. The resveratrol is well absorbed in humans as well. In a study conducted on human volunteers, resveratrol was administered orally to 15 volunteers at a 500 mg oral dose. The result revealed that 75% of resveratrol dose was absorbed [22].

It should be noted that the kind of food ingested and its lipid content has limited effect on resveratrol bioavailability. Trace amounts of trans-resveratrol were detected in blood samples after 30 min of red wine ingestion and resveratrol glucuronides were predominant after longer times [23]. It seems that health benefits of red wine consumption are due to the whole antioxidant active ingredients in red wine.

Once in the bloodstream, three different forms of resveratrol can mainly be found which are: glucuronide, sulfate or free form. The free form has the affinity to lipoprotein and albumin that it could be a natural polyphenolic reservoir, playing a role in its distribution and bioavailability. This binding enhanced in the presence of fatty acids [24].

Phase II metabolism of resveratrol occurs in the liver. In vivo, resveratrol is absorbed and rapidly metabolized to its stable glucuronides, sulfates and hydroxylates. Resveratrol has been indicated to be metabolized to its 3- and 4′-*O*-sulfate, and 3-*O*-glucuronide conjugates less than 2 h after ingestion in healthy human [25,26]. Bode and colleagues have shown that the fractional ratio of metabolite among the individual varies mainly due to intestinal microbial that plays a role in resveratrol metabolism [27]. The pharmacokinetic assessment of resveratrol in healthy volunteers displayed rapid and extensive metabolism to resveratrol-4′-Oglucuronide, resveratrol-3′-*O*-glucuronide and resveratrol-3-*O*-sulfate following the delivering of single or multiple oral doses between 0.5 to 5.0 g each [25,27]. Even with a hefty dose, this does not leave much opportunity for resveratrol to impart its therapeutic action.

Resveratrol is poorly bioavailable, and that considered the major hindrance to exert its therapeutic effect, especially for cancer management [20]. The bioavailability studies of resveratrol from three different sources (white wine, grape juice, vegetable homogenate) at lower doses (25 mg per healthy subject) demonstrate that the mean proportion of free resveratrol in plasma was 1.7–1.9% with a mean plasma concentration of free resveratrol around 20 nM [28]. Boocock and his colleagues studied the pharmacokinetic of resveratrol; in vitro data showed that minimum of 5 µmol/L resveratrol is essential for the chemopreventive effects to be elicited [29]. Despite the low bioavailability of resveratrol, it shows efficacy in vivo. This may be due to the conversion of both glucuronides and sulfate back to resveratrol in target organs such as the liver [21,30]. In addition, that could be due to the enterohepatic recirculation of metabolites, followed by their deconjugation and reabsorption [21].

Glucuronidation of the cis-form is five to ten times faster than the trans-form, thus causing a lower bioavailability of the cis-form [21]. Furthermore, there is some possible variation in individual response upon oral ingestion of resveratrol that could affect resveratrol bioavailability [31]. Such variation could be due to the difference in genetic background between different individuals [32]. A study was conducted on healthy volunteers to evaluate cancer chemopreventive properties of resveratrol. Results of this study suggest that repeated administration of high doses of resveratrol generates a higher plasma concentration of parent and a much higher concentration of sulfate and glucuronide conjugates in the plasma [33]. The doses tested in this study were 0.5, 1.0, 2.5 or 5.0 g daily for 29 days. No toxicity was detected, but moderate gastrointestinal symptoms were reported for 2.5 and 5.0 g doses [33].

They recently proposed some possible scenarios to increase resveratrol’s bioavailability by inhibiting resveratrol metabolism in vivo. Some of these scenarios based on the combinations of resveratrol with natural agents, nanoparticle-mediated delivery and use of naturally occurring or synthetic analogs of resveratrol [29].

The most useful way to enhance the efficacy of resveratrol is to increase the proportion of free resveratrol available at the target organ site. Regarding this, other components, preferably naturally occurring agents, are using to delay the rapid metabolic and elimination of resveratrol. At present, the co-administration of piperine with resveratrol was used to enhance resveratrol bioavailability by several private industries. Wightman and colleagues showed that piperine could enhance the bioefficacy of resveratrol when co-supplemented in healthy human volunteers [34]. In an In vitro study, co-supplementation of resveratrol with quercetin, as both polyphenols co-exist in red grapes, and other plants, resulted in inhibition of resveratrol glucuronidation and sulfation in the liver and intestine. On the other hand, in vivo results showed no inhibitory effect of quercetin [35,36]. Basu and colleagues believed that the use of curcumin would inhibit the glucuronidation of resveratrol in mice [37].

The rapid metabolism of resveratrol leads to produce five distinct metabolites, which are resveratrol monosulfate, two isomeric forms of resveratrol monoglucuronide, dihydroresveratrol monosulfate and dihydroresveratrol monoglucuronide. Various in vitro and in vivo studies made by some researchers believe that resveratrol metabolites may have their different biological activity from that of free resveratrol. Interestingly, there is some evidence that resveratrol 3-sulfate could also afford chemopreventive effects [38]. Moreover, it is possible to assume that the anti-cancer effect of resveratrol is due to the synergistic effect of resveratrol and its metabolite. Table 1 shows the efficacy of a number of resveratrol metabolites.

## 3. Tumor Microenvironment

The tumor microenvironment is an extraordinarily dynamic and complex system of many cell types. Including smooth muscle cells, endothelial cells, fibroblasts of various phenotypes, myofibroblasts, mast cells, T cells, B cells, natural killer, neutrophils, granulocyte and antigen-presenting cells such as macrophages and dendritic cells. All these cells are involved in tumor progression [49].

Realizing the importance of the tumor’s microenvironment in the development of cancer has led to a shift from a view of cancer development to the concept of a complex tumor ecosystem. In which cellular and molecular components are as active as the cancer cells themselves for the development of cancer and metastases [50]. One feature of all ecosystems is that slight modifications in one component may cause a significant reorganization of the entire system. As a result, interference with any component of the tumor ecosystem provides an opportunity to counteract the development of cancer [50].

Despite a growing understanding of cancer cell transformation, current treatments are not complete or are transiently useful for most types of cancer. Although genetic changes in tumor cells are necessary for tumor progression, they are not sufficient to give cancerous cells malignant properties. Different types of host cells are involved in tumor infiltration (Figure 2); this leads to the development of a stromal compartment mixed with tumor cells. Such microenvironment is required by cancer cells to generate a permissive environment for the invasion of epi-genetically modified tumor cells [51].

The importance of tumor microenvironment (TME) in cancer progression is reflected by the fact that tumor local invasion and metastasis are a result of collaborative interactions between neoplastic cancer cells and stromal cells [52]. Thus, in the past decade, TME and its “stromal” cell studies have increased dramatically, and they are now embracing a wide range of investigations [53,54].

The stromal component of the TME grouped into three general classes: infiltrating immune cells (IICs), angiogenic vascular cells (AVCs) and cancer-associated fibroblastic cells (CAFs). These cells are involved in seven cancer hallmarks including sustaining proliferative signaling, evading growth suppressors, resisting cell death, enabling replicative immortality, inducing angiogenesis, evading immune destruction and activating invasion and metastasis [52].

Therefore, this highlights the realization that cancer cells, despite all of their mutation benefits, do not act alone in detailing the disease. A better understanding of how the tumor environment affects the development of cancer must be obtained. This provides us new goals to isolate and destroy cancer cells by interfering with the complex crosstalk between cancer cells, host cells and the extracellular matrix surrounding them.

## 4. Targets of Resveratrol in Tumor Microenvironment

### 4.1. Reactive Oxygen Species (ROS)

Reactive oxygen species (ROS) are reactive chemical species containing oxygen, such as superoxide and hydrogen peroxide. ROS are produced naturally as by-products of normal aerobic metabolism. ROS levels can increase dramatically during times of environmental stress, resulting in significant damage to cell structures, and this occurs in a condition known as oxidative stress. At low levels, ROS plays a critical role in cell signaling processes. At a higher level, it induces apoptosis [55].

It has been shown that ROS mediates post-translation modifications of p53 (a tumor suppressor gene) and urges to disable the permeability of the mitochondrial membrane and fragmentation of nuclear DNA [56,57]. These studies suggest the novel function of ROS as p53 activators or p53 downstream mediators. P53 is highly expressed by post-translational modifications, including phosphorylation, acetylation and glycosylation, when cells are exposed to oxidative stress [58,59]. These events occur fast and cause the activation of p53, resulting in either apoptosis or cell cycle arrest.

Recent studies have shown that resveratrol increases ROS generation and decreases mitochondrial membrane potential [60,61]. In one study, A375SM melanoma cells were treated with resveratrol. This treatment decreased the viability of melanoma cells by activating the expression of both p21 and p27, which promoted cell cycle arrest. Furthermore, resveratrol increased the expression of cellular ROS [62]. Various human cancer cell lines (MCF-7, MDA-MB-231 and H1299 cells) exhibited an increase in the ROS levels after resveratrol treatment [63]. Additionally, the use of resveratrol with cisplatin in malignant human mesothelioma cells (MSTO-211H and H-2452 cells) synergistically induces cell death by increasing the intracellular ROS level [64]. Resveratrol decreases autophagic flux, and elevate the ROS level resulting in cell death in human cholangiocarcinoma cells [65]. Therefore, resveratrol could manage the ROS level in a specific manner.

On the other hand, resveratrol exhibits anti-oxidant properties that have been shown to be related to its chemopreventive effect. This effect is mediated by its ability to scavenge oxygen radicals, [66].

It seems that resveratrol has two contradictory effects with oxidative stress. In its therapeutic effect, it increases oxidative stress to inhibit cancer cells. In its chemopreventive effect, it acts as scavenger of reactive oxygen species to protect cell from mutations.

### 4.2. Tumor Associated Macrophages and Indoleamine 2,3-Dioxygenase in Dendritic Cells

Different innate immune cells were observed to regulate tumor microenvironment through stimulation of angiogenesis, induction of fibrosis and immune system suppression [67]. Among these cells are macrophages which present in two phenotypes M1 and M2. M1 is the physiologically active phenotype and produce different cytokines such as IL-12, IL-1, IL-8, IL-15, IL-18, IL-23 and TNF-α. These cytokines are produced in response to binding and recognition of microbial molecular pattern [68]. The activity of M1 macrophages, in particular IL-12 production, induce a shift in the immune system toward Th1 response which is important to eliminate cancer cells [69]. On the other hand, tumor infiltrating macrophages (M2 phenotype) produce IL-10 and transforming growth factor (TGF-β). These cytokines shift T cell differentiation away from Th1 and induce immune suppression [70,71]. TGF-β also induces tissue fibrosis by activating fibroblasts and other mesenchymal cells [72]. Another mechanism for tumor associated macrophages (TAM) to stimulate tumorigenesis is by inducing tumors cells to express immune checkpoints. Programmed death-ligand 1 (PD-L1) is one of the immune checkpoints that can be stimulated by TAM. Engagement of PD-L1 with its receptor on T cells inhibits the ability of these cells to recognize tumor antigens [73].

Treatment of lung cancer cells with intra-peritoneal injection of resveratrol (100 mg/kg) suppressed M2 polarization and caused tumor regression in vitro and in vivo. This inhibition was associated with a decreased activity of signal transducer and activator of transcription 3 (STAT3) [74]. In another study, inhibition of M2 macrophage activation and differentiation was observed after oral administration of resveratrol at a concentration of 25 mg/kg, twice daily. This treatment inhibited osteosarcoma metastasis to the lung and liver and caused lymphangiogenesis inhibition [75]. Repolarization of TAM from M2 to M1 was also achieved using a combination consisting of resveratrol, curcumin and epicatechin gallate. This combination also induced a decrease in IL-12 which resulted in intra-tumor infiltration of natural killer and cytotoxic T lymphocytes [76]. HS-1793 is a synthetic analog of resveratrol with improved antitumor effects. Treatment of breast cancer bearing mice with 1.5 mg/kg of HS-1793 (twice a week for 30 days) resulted in tumor regression. This regression is mediated by induction of IFN-γ production and reprogramming M2 macrophages toward M1 phenotype [77].

Dendritic cells are the principal antigen presenting cells in the immune system and involved in the immune regulation of cancer [78]. In addition to its role in inducing antitumor immune response, dendritic cells also involved in immune tolerance [79]. The immune tolerance induced by dendritic cells is mediated by different mechanisms. One of these mechanisms is the activity of the enzyme indoleamine 2,3-dioxygenase (IDO) which is responsible for degradation of the essential amino acid tryptophan and the generation of kynurenine [80]. Dendritic cells with active IDO were detected in tumor microenvironment and draining lymph nodes [81]. These cells exhibit immune suppressive effects mediated by different mechanisms including T lymphocytes and NK cells inhibition and promotion of regulatory T cell differentiation [82]. Another immune suppressive mechanism of IDO harboring dendritic cells is through rapid depletion of tryptophan. Low tryptophan in tumor microenvironment initiates stress signals in T cells inducing anergy tryptophan in tumor microenvironment initiates stress signals in T cells inducing anergy in cytotoxic T cells and stimulating the differentiation of helper T cells to regulatory cells [83]. Such changes help in establishing immune suppressive microenvironment and support tumor growth.

Treatment of EG7 thymoma-bearing mice with resveratrol (50 mg/kg every 2 days for 3 weeks) caused tumor regression through the inhibition of IDO expression and activity. This inhibition resulted in regulation of cytotxic T cell polarization to enhance its antitumor activity [84]. On the other hand, an increase in the activity of IDO was observed after oral administration of 5 g resveratrol by 8 volunteers. The increase in IDO activity was significant after 2.5 and 5 h [85]. The contradictory effects of resveratrol on IDO deserve further research to clearly understand the signaling pathway involved in this regulation.

We can summarize the effects of resveratrol on macrophages and dendritic cells in tumor microenvironment to include the following: inhibition of M2 polarization of macrophage, inhibition of M2 macrophage activation, activates repolarization of macrophage from M2 to M1 phenotype, and inhibition of IDO expression and activity in dendritic cells.

### 4.3. Vascular Endothelial Growth Factor (VEGF)

Angiogenesis is a physiological or pathological process defined as the sprouting and remodeling of small new capillaries from the pre-existing blood vasculature, while vasculogenesis includes the migration, differentiation and association of endothelial precursor cells to form primitive blood vessels. The family of vascular endothelial growth factor (VEGF) has been known to have an important role in the homeostasis and development of the blood vessels as well as in lymphatic vessel formation. Multiple studies have shown that VEGF is the most efficient stimulator of angiogenesis [86,87].

The VEGF family of growth factors includes five members: the VEGF-A, B, C, D as well as placental growth factor (PlGF) [88]. Amongst these five members, VEGF-A is the most renowned as it is the major driver of angiogenesis and vasculogenesis, and it is commonly regarded as VEGF only [89]. VEGF-A, the first discovered member, is the most investigated one [89]. Physiologically, VEGF-A promotes the angiogenesis during embryonic development and is needed for tissue repair. However, in cancer patients, the VEGF-A production by the tumor results in “angiogenic switch” enhancing tumor growth and metastasis [87]. Recently VEGFs are identified as an important factor in tumor-induced immunosuppression and play a major role in tumor vessel growth (neoangiogenesis). Specifically, VEGF-A limits T-cell recruitment into tumors, enhances T-cell exhaustion, in addition to enhancing the immunosuppressive cells’ proliferation [90].

About 12 years ago the first evidence was published about the tumor-derived VEGF-A that might also affect cells other than endothelial and tumor cells, particularly some immune cells [91]. It now appears that proangiogenic growth factors such as VEGF-A may be involved in tumor escape from antitumor immunity [90]. Tumor vessels differ from their normal counterparts by their dynamic properties. Unlike normal blood vessels, they are always active, thus allowing ongoing tumor development and production of new vessels. The importance of neovascularization in tumor growth depends on the tumor type [90].

Hypoxia results from the unavailability of oxygen due to increased tumor mass producing proangiogenic factors such as VEGF-A, mediated by the transcription factor called hypoxia-inducible factor (HIF)-1. The new vasculature formed under the influence of VEGF enables the tumor to meet its nutrient requirements. However, tumor vessels are both structurally and functionally abnormal, immature, irregularly shaped, dilated but complex [87]. The tumor vascular network is also prone to hemorrhage and necrosis, because of an increase in interstitial fluid pressure related to overproduction of VEGF-A (or vascular permeability factor) [90].

Two types of receptor tyrosine kinases regulate the function of VEGF on endothelial cells. They are the VEGF receptor-1 (VEGFR-1) and VEGFR-2. Both HIF-1a and VEGF are linked to aggressive tumor types and usually are over-expressed in numerous types of tumors as well as their metastases in human [92].

Controlling cancer growth by signal-blocking of VEGF-mediated responses is imperative and certain drugs were developed for this purpose. The categories of these drugs that antagonize the effect of receptor tyrosine kinases in oncological studies have been developed. Antibodies that were developed against VEGF are those targeting VEGFR-2, in addition to small molecular inhibitors targeting VEGFR-2 kinase domain [92]. The innovation of inhibitors blocking the signaling pathway of VEGF/VEGFR2 could be encouraging novel method in angiogenesis-related diseases’ treatment.

Resveratrol was investigated to withhold the stimulation of numerous transcription factors that inhibit the action of protein kinases. This investigation also covers the down-regulation of the products of the following genes, COX-2, 5-LOX, VEGF, IL-1, IL-6, IL-8, AR and PSA [93]. This recent study also emphasized and defined the pharmacological significance of the activity of resveratrol and ginkgetin in angiogenesis, in cancer cells and its related signaling mechanisms [94]. It was found out that combining resveratrol and ginkgetin can be efficient as an angiogenesis inhibitor and has the potential characteristics of a drug for cancer treatment and drug development. This combination was found to suppress VEGF through a succeeding steps interfering with the VEGF-related signaling transduction. These activities of resveratrol and ginkgetin in comparison with other anti-angiogenesis agents, can be studied further for the treatment and prophylaxis of different types of angiogenesis-related illnesses and can be a novel angiogenesis inhibitors [94].

Lin et al. [95] investigated whether resveratrol at a dose of 1 or 2.5 µM, can efficiently break VEGF-mediated tyrosine phosphorylation of the vascular endothelial (VE)-cadherin in addition to its partner, β-catenin in Human umbilical vein endothelial cells (HUVECs). As established by immunofluorescence, the retention of VE-cadherin at cell–cell contacts reflected the inhibitory effects of resveratrol. The investigation proved that VEGF stimulated a rise in the amount of peroxide, that effectively inhibited by resveratrol. The disruption of ROS-dependent Src kinase activation and the following VE-cadherin tyrosine phosphorylation was related to resveratrol inhibition of VEGF-induced angiogenesis.

Resveratrol inhibited the expression of some molecules and thus far can be considered an effective anti-cancer therapy for cancer and for the prevention of its metastasis [96,97,98]. Resveratrol has a notable anti-angiogenic activity on the growth of DENA-induced hepatocellular carcinogenesis through blocking VEGF expression via regulating HIF-1a [99]. Thus, by preventing also the activation of the MAPK and PI3K/Akt signaling pathways, it suppresses HIF-1a and VEGF release in ovarian cancer cells of humans [100].

By preventing tumor progression through limitation of tumor-induced angiogenesis, anti-angiogenic agents have shown effectiveness in patients with a variety of solid tumors. Alteration of the tumor-induced immunosuppressive microenvironment was the target of VEGF-targeting therapies and this was widely done by the enhancing Th1-type T-cell responses as well as increasing tumor infiltration by T cells [90]. Its effectiveness and properties shed light for cancer management, especially those strategies that merge antiangiogenic agents with immunotherapy. Early clinical studies of these combined approaches in addition to pre-clinical models provide promising results.

Treatments which prevent angiongenesis have manifested efficacy among patients with solid tumors as these treatments have helped to prevent the progression of tumors by preventing angiogenesis. Treatments which focus on the VEGF have been noted to impact on immunosuppressive setting, promoting Th1-type T-cell reactions and facilitative access by T cells to the tumor [90]. VEGF targeting interventions modulate immune functions which then allow for new avenues in cancer management, especially using treatments which use anti-angiogenic drugs and therapies which improve immune functions. Favorable results were seen in preclinical studies including those which have evaluated these combined therapies.

In summary the anti-angiogenic activity of resveratrol is mediated by two main mechanisms. The direct mechanism involves suppressing VEGF production by inhibiting the signaling pathway involved in its synthesis. The indirect method is through inhibiting the production of HIF-1 which results in preventing VEGF release.

### 4.4. Fibrosis

Not surprisingly, tumor aggression and poor patient prognosis correlate with degree of tissue fibrosis in addition to the level of stromal stiffness [101]. Fibrosis promoting malignancy is a result of extracellular matrix (ECM) remodeling, deposition and cross-linking. This process characterized the development of a tumor. The stiffened stroma also promotes cell growth, survival and migration that drive mesenchymal transition. A stiff ECM also induces angiogenesis, hypoxia and compromises anti-tumor immunity. Tissue fibrosis is frequently noticed in the tumor microenvironment related with rapid fibroblasts’ proliferation [67].

Fibroblasts on the other hand secrete various cytokines and chemokines including TGF-β, IL-1, IL-6, IL-8, CXCR4, CXCL12 and monocyte-chemotactic protein 1 (MCP-1) [102]. They also produce platelet-derived growth factor (PDGF), HGF, stromal-cell-derived factor 1 (SDF1), VEGF and basic fibroblast growth factor (bFGF) [103] which promotes tumor growth.

In patients with various types of cancers, fibroblasts can pose a worse clinical outcome [104,105,106]. More tumor activities are linked to cancer-associated fibroblasts (CAF) because of the secretion of MMPs which is responsible for the ECM breakdown that enhances cancer cells to leak into the vasculature and metastasize to distant sites [103]. A certain process that signals the re-differentiation of cancer cells of epithelial origin into mesenchymal cells with properties of stemness (the presence of a stem cell-like phenotype in tumors), was associated with MMPs inducing the transition from epithelial to mesenchymal form (EMT). At this point, the EMT serves as biomarker of poor prognosis [104]. These findings also revealed the potential of fibroblasts inhibition as an effective means for treating cancers.

Still, there were previous studies which indicated how fibroblasts seem to have a greater role in the development of tumors. One of these studies [107] indicated that focusing on the fibroblast activating protein (FAP) would not lead to the regression of the tumor but would instead cause bone marrow toxicity. In the study by Özdemir and co-workers [108] where the smooth muscle actin positive myofibroblasts linked to the early stages of pancreatic cancer is deleted, the tumor phenotype appears to be more aggressive and poorly differentiated. In effect, in the development of the extra-cellular matrix and stimulating fibrosis in developing tumors, fibroblasts as well as myofibroblasts are very much important.

Recently, resveratrol has been used in several complementary and alternative approaches for treating cancer [109]. The activation of pancreatic stellate cells (PSCs) is a crucial step in the progress of pancreatic fibrogenesis that leads to chronic pancreatitis and pancreatic cancer. Resveratrol was found to effectively impede the activation, invasion, migration and glycolysis of PSCs induced by reactive oxygen species (ROS) by down-regulating the expression of microRNA 21 (miR-21) and also by increasing the phosphatise and tensin homolog (PTEN) protein levels [110]. In the same study, the results further demonstrated that resveratrol inhibited the invasion and migration of pancreatic cancer by suppressing ROS/miR-21 mediated activation and glycolysis in PSCs. In addition, resveratrol was shown to suppress the PSCs viability by reducing several major fibrogenic mediators, such as a-SMA, type I collagen and fibronectin, which are associated with regulation of the NF-kB signaling pathway [111]. Additional studies are needed about the use of phytomedicines like resveratrol, as a new anti-fibrotic agent.

### 4.5. Interleukin-6 (IL-6)

A pleiotropic protein that is a pro-inflammatory cytokine called IL-6 was involved in biological activities, involving cancer and autoimmune diseases in addition to other processes. It has 21–28 kDa 4-helix bundled glycoprotein consisting of 184 amino acids [112,113,114]. Normal cells like macrophages, monocytes, as well as stromal, hematopoietic, muscle and epithelial cells produced IL-6. This protein was found to play a crucial role in immunity, inflammation, metabolism, reproduction, hematopoiesis, bone remodeling, neural development and angiogenesis [115,116].

In a tumor microenvironment, in addition to the tumor cells themselves, various types of cells, which include tumor infiltrating immune cells, stromal cells and fibroblast, produced IL-6, and several factors can initiate this process [113,114,117,118,119]. High levels of IL-6 enable the tumor cells to become resistant to both cancer immunotherapy and chemotherapy by increasing an anti-apoptotic and proliferative state in tumor cells [120,121,122].

A systematic review about IL-6 revealed that it was increased in serum or plasma in patients with gastrointestinal tract (GIT) cancer compared to healthy individuals [123]. This systematic review evaluated the diagnostic value of IL-6 to find possible methods for early detection of GIT cancers. IL-6 is not cancer-specific and it has insignificant value to be considered as independent single diagnostic biomarker. This review found several information about IL-6 protein including 48 publications that investigate IL-6 as a cancer biomarker in 5316 individuals with GIT cancer [123].

Majority of the studies that were also conducted demonstrated that for patient with indefinite cancer symptoms IL-6 might be used as a bed-side test since it correlated with the clinical characteristics of the patient. This implied that the use of IL-6 as a predictive biomarker for patients with gastric and pancreatic cancer, in addition to those with bile duct cancer, can be effective and promising. However, the use of IL-6 in colorectal cancer was still unsafe and too ambiguous to deduce its effectiveness. Although studies about cancer cells showed remarkable correlation between IL-6, chemoresistance, and increased proliferation, as well as IL-6 to be used as a therapeutic target, they are all currently being investigated. Currently, the use IL-6 in improving the quality of life and in decreasing severe cancer symptoms like fatigue and cachexia in patients with GIT cancer are subjects for further research [123].

IL-6 stimulation that lead to increased androgen receptor (AR) transcriptional activity as modulated by resveratrol was the concern of previous studies [124]. The study revealed for the first time that stimulating human prostatic carcinoma cell line- fast growing colony (LNCaP-FGC) cells with IL-6 clearly induced the AR transcriptional activity while resveratrol reduced IL-6-induced AR activity. Moreover, higher levels of AR activity were elicited when cells were incubated with a combination of IL-6 and dihydrotestosterone (DHT) as compared to those treated with DHT or IL-6 only. Not only does resveratrol inhibit DHT- or IL-6-induced AR activity, but it also reduced the AR transcriptional activation as well as the production of prostate-specific antigen (PSA) in cells stimulated with DHT and IL-6 [124]. The study was able to promote new perceptions on the function of IL-6 in AR activation in prostate cancer cells and as inhibited by resveratrol. This indicated a new molecular feature of chemopreventive effects in prostate cancer.

Androgen levels were also noted to have increased due to IL-6 actions on LNCaP cells [125]. This seems to also cause IL-6-stimulated AR activity. As IL-6 impact on cancer cells increased serine-threonine kinase Pim1, over stimulation also led to AR transcriptional reactivity [126]. Moreover, with the phosphorylation on AR caused by Pim1 on serine-213 residue, cancer cells (prostate) have become more aggressive and unaffected by androgen ablation therapy [127]. As such, Pim1 stimulation may be the cause for the IL-6 AR activity seen in LNCaP-FGC cells. It may be interesting to study the impact of resveratrol on Pim1 in LNCaP cells. The actions of Pim1 would be impacted by STAT3, leading to the possibility that IL-6-mediated STAT3 activity may cause Pim1 action, which can also cause AR stimulation [128]. The strong chemical elements affecting Il-6-induced AR reactions and the cellular structure supporting cross-functions in STAT3 and AR reactions have to be studied at more length. Table 2 Summarized targets for resveratrol in tumor microenvironment and the mechanisms of action.

## 5. Resveratrol in Anti-Cancer Clinical Studies

### 5.1. Beneficial Effects of Resveratrol and Its Pharmaceutics

It is apparent that resveratrol shows efficient results towards cancer and towards inflammation. However, these studies were mostly carried out in vitro and on animal subjects. It is important to study how resveratrol would affect humans especially as results on animal models cannot be used to draw conclusions on human usage especially as the anatomy and physiology of humans are different from animals. The use of resveratrol for cancer and inflammatory management in humans is very much limited, but there are some studies associated with resveratrol usage for cardiovascular purposes. Such application has been noted due to the newness of this treatment in relation to cancer and inflammation. More studies are now being carried out to further establish how resveratrol would affect cancer and inflammation among humans.

The US National Cancer Institute and the UK Medical Research Council supported a particular study to investigate the pharmacokinetics of resveratrol and the first phase of its clinical trials was completed [25]. The study aimed to evaluate the safety and plasma levels of resveratrol in healthy individuals when orally administered in single doses of 0.5, 1, 2.5 and 5 g. Forty healthy volunteers with age range of 19–61 years were given a single oral dose of up to 5 g resveratrol. After the trials were completed, it was observed that resveratrol is present in urine and plasma of subjects in lower concentration after absorption as compared to the predetermined cancer chemopreventive efficiency level of at least 5 µmol/L [25]. The study revealed that the dose given was unsuccessful in producing any significant side effects [25]. With this study, it was also observed that chemoprevention-related pharmacological effects was elicited with this minimum level of resveratrol compared to the data from other various in vitro studies where rapid bioconversion of resveratrol into its metabolites was present in the urine of the subjects in high concentrations. This rapid conversion resulted in low plasma and urine concentrations. This study [25] suggested that it would be advantageous to conduct further investigation to verify if target tissues are capable of converting resveratrol metabolites into the putatively active parent molecules. There is also a need to look to metabolites of resveratrol such as monoglucoronide and sulfate with plasma concentrations of 0.9–4.3 and 4–14 µmol/L, respectively as possible sources of chemopreventive activity that was observed in the parent molecule to support previous similar recommendations [129,130]. Interpretation of in vivo results obtained in rat models can only be interpreted through studying the rapid metabolism of resveratrol.

According to a previous study, after an oral dose of 25 mg/70 kg resveratrol administered to healthy human participants, the compound predominantly appeared in the form of glucuronide and sulfate conjugates in serum and urine and reached its peak concentrations in serum about 30 min after ingestion [28].

The clinicaltrials.gov site that provided the database of supported human clinical trials, showed that there are roughly 80 resveratrol studies. Several trials were conducted to evaluate the safety, pharmacokinetics and bioavailability of resveratrol and merely limited recorded trials focused on evaluating the effectiveness of resveratrol in certain cancers.

Table 3 presents the summary of the completed human trials of resveratrol. Various forms of resveratrol, including trans-resveratrol, extract of *Polygonum cuspidatum* (Japanese knot-weed), SRT501 (micronized resveratrol), resveratrol rich seedless red grapes/grape juices (Muscadine grapes) and microencapsulated resveratrol were used in the trials. Although majority of the trials conducted focused on the evaluation of colorectal cancers, it included various cancer types like multiple myeloma, breast cancer, follicular lymphoma and neuroendocrine tumors. As a result of potentially direct contact and prolonged exposure of resveratrol to colonic tissue, the research study however found out that resveratrol has proven to be mildly effective in colon cancer clinical trials. Additionally, the efficiency of the intestinal epithelium in the absorption of nutrients and active molecules from food and food components should also be considered.

### 5.2. Completed Clinical Studies

Still, few clinical studies on the anti-cancer property of resveratrol were conducted in vitro for humans although there were enough available data of clinical trials conducted on animals proving its effectiveness as anti-cancer agent. Studies on the effects in humans were also partly limited where most of the studies conducted were done in cell-cultures and pre-clinical models. As the results from testing animals cannot be regarded as true for humans because of genetic variations and differences in metabolic profile, the physiological effects of the drug were also analyzed and assessed for humans.

A clinical trial to check the impact of resveratrol on cancer cells was first undertaken by Nguyen et al. [131]. They compared the use of low-dose plant-based resveratrol with freeze-dried grape power resveratrol on colon cancer patients, specifically Wnt signaling. The authors noted that the inhibition in Wnt gene expression was significant in cases of normal colon mucosa; but there were no changes seen in relation to the cancer cells [131]. In the same patient, cancer tissues and colonic mucosa were secured at the same time in the same operation. Similar preparations and analysis were also carried out. It is surprising to note how patients undergoing resveratrol treatment experienced major changes in their composite Wnt gene levels in cases of colon cancer. There were no major changes for some genes (myc and cyclin D1) following resveratrol use. However, these were heightened in the case of cancer tissues following resveratrol grape powder use. This was however a pilot study and only few participants took part in the study. The study was meant to check biomarkers in relation to target genes (Wnt). Some issues noted in this trial was noted in the fact that the researchers were informed that the plant based capsules had 20 mg resveratrol but the HPLC assessment noted only 3.886 mg of resveratrol in each capsule [131]. Moreover, there was also quercetin detected in the capsules. The clinical trial highlighted the value of securing the correct dose in resveratrol for use in future studies. Another issue noted was associated with the quality control applied in the trial. The sample has to be assessed for safety and must not experience any interference.

The presence of resveratrol and its metabolites in human tissues was reported by Patel and his colleagues [132]. The level of resveratrol was successfully identified and quantified as well as its metabolites which are resveratrol-3-*O*-glucuronide, resveratrol-4′-*O*-glucuronide, resveratrol-3-*O*-sulfate, resveratrol-4′-*O*-sulfate, resveratrol sulfate glucuronide and resveratrol disulfate by HPLC/UV found in operated colorectal tissue. The study was made to determine if 0.5 and 1.0 g doses of resveratrol can reduce the proliferation of tumor cells by 5% in all the colon cancer patients [132] The authors have suggested from this study that the doses administered to colorectal tissues of patients were enough to induce chemopreventive activities although variable concentrations were used. Quite surprisingly, higher levels of resveratrol and its metabolites were obtained from tissues originating from the right side of the colon as compared to those originating from the left. The effects on the prognosis of colorectal cancer with these findings were still subject to investigations as the high variability of resveratrol metabolites present in tissues should be taken into consideration in this drug for clinical trials.

Higher levels of resveratrol in plasma and in hepatic tissue was detected after 5 g administration of SRT501 in patients with colorectal cancer and hepatic metastasis who scheduled to undergo hepatectomy as reported by Howells and his colleagues [133]. Micronized resveratrol was found to be tolerated by patients more readily with mild adverse events compared to non-micronized resveratrol. The concentrations of resveratrol that were recovered from hepatic metastases were enough to elicit pharmacologic effect. Significant rise in the level found in tissues were noted. This was particularly higher compared to the level found in tissues of subjects taking placebo [133]. On the other hand, there was no significant change found in other tested biomarkers including AKT1, GSK-3, survivin and PARP. Generally, SRT501 micronized suspensions were tolerated more readily and with superior bioavailability compared to nonmicronized resveratrol [25,33]. Increasing the doses was recommenced to achieve significant apoptosis stimulation.

Assessments of safety, pharmacokinetics and effectiveness of SRT501 alone or combined with bortezomib were studied by Popat et al. through a series of clinical trials. This was conducted in relation to the study of the effectiveness of a currently approved chemotherapy drugs for patients with multiple myeloma. From this study, a series of unexpected adverse events including renal toxicity were reported [134]. The trial was then immediately terminated because of the occurrence of severe adverse events like nausea, diarrhea, vomiting, fatigue, anemia, infections and serious renal failure. Renal failures were evident only in patients with multiple myeloma. This can be attributed to the fact that renal impairment is quite common with 50% probability in myeloma patients. The trial was conducted in specific populations and highlighted the risks of occurrence of resveratrol (and possibly other agents). Drug interaction and occurrence of secondary complications related to each disease should be carefully evaluated when conducting resveratrol-based clinical trials. In this case, it appeared that the increased risk of kidney problems in multiple myeloma patients can be further aggravated by the treatments itself. The reason behind this comprehensive interaction at a molecular level has not been identified and studied. However, determining the exact and comprehensive mechanism of action of resveratrol in a particular cancer type may be useful for succeeding research.

Zhu and his colleagues conducted another study that assessed the effects of resveratrol on methylation of certain breast cancer-related proteins in women with increased risk. The study included women with relative in the first degree with breast cancer and with a Gail risk of >1.66% in developing breast cancer. The study also included women with breast biopsy demonstrating atypical hyperplasia in situ, and those with invasive breast cancer (previously diagnosed but currently free of disease) [136]. The study was a comparison of the effects of either 5 or 50 mg of trans-resveratrol twice per day (for 12 weeks) on methylation of certain genes, as compared to those taking placebo. The study demonstrated increased levels of trans-resveratrol and resveratrol-glucuronide in circulation and eventually the reduced methylation of RASSF-1a, together with decreasing prostaglandin E2 (PGE2) expression in the breast [130]. The disease progression from pre-cancer to cancer in the breast was associated with the increased levels of PGE2 and methylation of RASSF-1a [137,138]. To determine whether abrupt or continuous supplementation of resveratrol to individuals with a higher risks of breast cancer will be needed, and whether a longer study might be required, different parameters were used. The study also emphasized careful determination of doses and treatment programs for each type of cancer or cancer-risk group.

Brown and colleagues noted how a major decline in circulating insulin-like growth factor (IGF)-I as well as IGF-binding proteins (IGFBP-3) among healthy individuals can be credited to the intake of resveratrol [33]. Any changes on the IGF axis arising from IGF-I or IGFBP-3 imply valuable data on the use of these proteins as possible markers to measure chemo-effective impact towards cancer. The study was able to note how IGF-I and IGFBP3 levels are associated with risks for possible diseases including cancers [139].

Brown and colleagues [33] established how the use of high dose (4000 mg/patient) pulverized muscadine grape skin was considered safe for use among patients with recurring prostate cancer. This was with ellagic acid, quercetin, as well as resveratrol. However, more studies are needed to determine best therapeutic dose [126]. In another similar study involving placebo trials with two doses of resveratrol spanning four months, lower androstenedione, dehydroepiandrosteron and dehydroepiandrosteron-sulfate in the blood was registered even as the size of the prostate was not changed among those with benign prostate hyperplasia [140].

With the use of 5 mg/day (six days) of resveratrol supplements, protein carbony levels increased including cytoprotective enzyme NQO1 as seen in the patients with colorectal cancer. This was not seen in control subjects [141]. Some results however indicate how resveratrol supplements may have negative effects on cancer patients even with its proven benefits. In the second phase of a clinical study evaluating patients with multiple myeloma undergoing SRT501 supplements daily, negative effects were noted. Some of these effects include kidney toxicity which unfortunately led to a patient’s death [134]. Still, SRT501 at high dose was registered at safe in affected populations following the conduct of other clinical studies [133,134].

Available data on human trials to assess the effectiveness of resveratrol in cancer treatment were still very few. The data exhibited unpredictable outcomes in the use of resveratrol since most of these clinical trials used small sample size of patient as well as different doses and routes of administration.

## 6. Resveratrol in the Pharmaceutical Industry and Supplement Market

Supplement markets were now enticed to develop resveratrol drugs because of its health promoting potential. The Biotivia Longevity Bioceuticals Company published that they have developed a micro-encapsulated formulation Transmax TR that will protect resveratrol from stomach acid to be release safely into the blood stream. Likewise, revgenetics Micronized Trans-Resveratrol powder can be absorbed up to 220% better than regular resveratrol supplements. Another branded formulation of resveratrol was named Microactive^®^ and Resveratrol SR and manufactured by Bioactives. This compound is capable of sustained release for over 12 h to increase intestinal residence time. Moreover, a pilot study posted on the website of the manufacturing company stated that the administration of Microactive^®^ Resveratrol SR capsules after breakfast to healthy individuals enable the drug to stay in the bloodstream for a longer period of time in contrast to the effect of regular resveratrol formulation. However, limited information was available regarding its formulations and clinical testing conducted in the mainstream of several scientific literature [142]. One of the few papers available contained a pilot study conducted by Howells and colleagues has assessed the safety, pharmacokinetics and pharmacodynamics of SRT501, a micronized form of resveratrol designed by Sirtris, a GSK Company. The study revealed that in tissues with higher levels of plasma resveratrol, SRT501 was tolerable and noticeable in hepatic tissue as compared to the data from previous studies. It was also found out that the time required to reach the highest level of resveratrol was also longer in the SRT501-administered patients. A dose of 5 g daily for 14 days was administered. The test was conducted in patients with colorectal cancer and hepatic metastases and who had been advised medically to undergo hepatectomy. An increased amount of cleaved caspase-3 within hepatic tissue with resveratrol administration was observed, suggesting that there was an increased apoptosis of cancerous tissue as compared to patients treated with a placebo [133]. Well-designed and effective clinical trials require the efforts of industries to support researchers in future studies. The industries’ financial support sets out our vision for the research to boost the productivity and the quality of the future studies. Partnership is at the heart of the successful approach. Researchers continue to recognize new challenges and technologies in resveratrol research and development grows substantially. A successful industries’ financial support needs to combine quickness with endurance. Researchers need apparent, trusted and consistent policies and direction, in addition to being flexible to change. Therefore, it is imperative that this industrial approach is continuously being updated and informed by evidence.

## 7. Resveratrol and Cancer Prevention

Given its ability to inhibit proliferation of different types of tumor cells in vitro, it is now understood that resveratrol plays an important role in cancer prevention. These in vitro effects have led to numerous studies to evaluate the potential of resveratrol use for patients who have a high risk for cancer. A decrease in DNA methylation in the tumor-suppressor gene Ras association domain-containing protein 1 (RASSF1A) was seen in women’s breasts, specifically those who had higher risks for breast cancer following the use of resveratrol (50 mg twice a day for 12 weeks) [136]. There was also a notable rise in sex steroid hormone binding globulin (SHBG) levels. This was associated with a decline in breast cancer risks seen in resveratrol supplements (1 g/day for 12 weeks) [143]. This implies how the drug seems to produce efficient results related to estrogen metabolism. From this, it can be implies that for those who have an elevated risk for breast cancer due to obesity and overweight issues or postmenopausal stage, resveratrol can help reduce cancer risks [144]. In addition, in terms of the impact of resveratrol on decreasing the biomarkers for cancer, improved effects were noted. As reviewed by Renehan et al., insulin-like growth factor (IGF-1) in circulation and IGF-binding protein 3 were associated with increased risks for some cancers [145]. As established by Brown et al. [33], the use of resveratrol (2.5 g/day for 29 days) led to reduced levels of IGF-1 and IGFBP-3 among healthy subjects. The impact of resveratrol to reduce IGF-1 and IGFBP-3 in the blood indicated that the drug has an impact on cancer and the development of cancer. Additionally in the study by Chow et al. [146], resveratrol (1 g/day for four weeks) impacted on isoenzymes (phase I) as well as their second phase (detoxification). This seems to be associated with the activation of cancer including detoxification [146]. Still, such benefits may be considered negligible. Undeniably, however, the drug shows major benefits in terms of preventing and treating cancer. More studies and human trials are needed to improve understanding of the drug and its potential for treating cancer.

## 8. Safety Aspects of Resveratrol Treatment

According to Planas et al. [147], blood and tissue toxicity was noted in the daily intake of high dose resveratrol (20 mg/kg) among rats. Crowell et al. [148] demonstrated how no negative effects in rats came about following the use of resveratrol (300 mg/day) for four weeks. This supports above results in different studies. Such studies clearly show how resveratrol and phytochemicals can be safely used in chemotherapies without any negative effects experienced. In general, therefore, it seems that there is an agreement that the safety profile of resveratrol is pristine [149]. Interestingly, no toxicity in clinical trials on humans was noted for those undergoing very high resveratrol daily doses (like 5 g/day) [25,33]. A possible area of future research would be to investigate why adverse events were developed when resveratrol was combined with bortezomib in multiple myeloma patients [134].

## 9. Conclusions and Future Perspectives

The evidence from many experimental in vivo and in vitro studies and few clinical trials suggests that resveratrol has a great potential as a prophylactic and treatment of a different types of cancers by reducing the course of the several stages of carcinogenesis. Resveratrol can also efficiently employ its anti-cancer effects in combination with other chemotherapeutics agents. Resveratrol has a very good safety profile. However, there are many conflicting findings partly due to dosing regimen used. Taken together, the completed clinical trials suggest that the dosage regimen of resveratrol that could be used for colon cancer treatment is 20–120 mg/day for two weeks or 0.5–1 g/day for one week, while 5 g for two weeks for patients with colorectal cancer with a hepatic metastasis. On the other hand, it is encouraged to administer resveratrol for a longer period of time to prevent cancer: 1 g/day for 4–12 weeks. However, more research using controlled trials is needed to definitely determine the exact recommended dosage regimen for cancer treatment and prevention and to compare and assess the long-term effects of different dosage regimens. Therefore, future research should deal with resveratrol’s delivery systems formulations, variation of resveratrol metabolism and its possible interactions with other agents, as well as exploring more bioavailable analogs of resveratrol. A number of important challenges need to be considered before bringing resveratrol to the bedside due to its rapid metabolism leading to a limited bioavailability in humans.

## Figures and Tables

**Figure 1 molecules-25-04282-f001:**
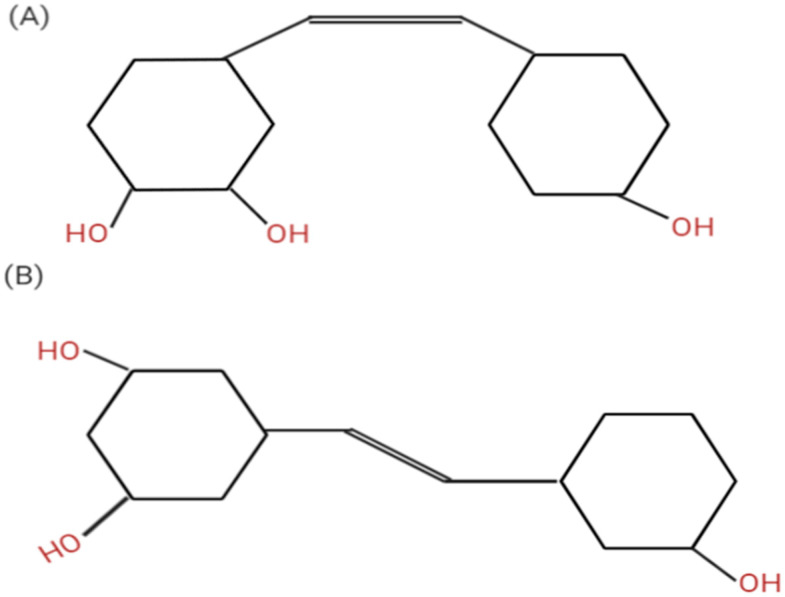
Resveratrol chemical structure in cis (**A**) and *trans* (**B**) isoforms. *Trans* is the biologically active form. Created using Biorender software.

**Figure 2 molecules-25-04282-f002:**
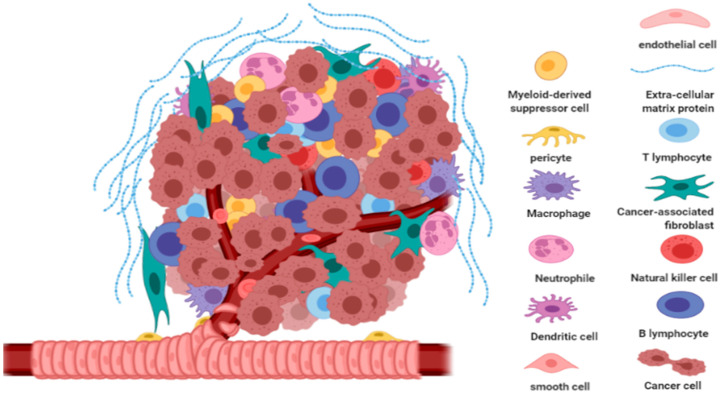
Tumor microenvironment. Various host cells and proteins work together to support cancer cell survival and progression. Created using Biorender software.

**Table 1 molecules-25-04282-t001:** Resveratrol metabolites and its pharmacological activities. (N = No reported activity).

Metabolites	Model of Experiment	Effect, Disease
Trans-resveratrol	Human [23,39]	antioxidant, cancer [23]
Trans-resveratrol-4′-*O*-glucuronide	Human [23,39] Mouse [40]	antioxidant, colon cancer [41] Delipidating age obesity [42]
Trans-resveratrol-3-*O*-glucuronide	Human [23,39]	Change lipid profile & control heart rate, Cardiovascular disease [43]
Trans-resveratrol-diglucuronide	Human [44] Mouse [40]	N
Trans-resveratrol-3-*O*-sulfate	Human [20,39] Mouse [40]	antioxidant, colon cancer [41]
Trans-resveratrol-4′-*O*-sulfate	Human [20,39] Rat [45]	antitumor, breast cancer [46]
Cis-resveratrol-3-*O*-sulfate	Rat [45]	change lipid profile, cardiovascular [43]
Trans-resveratrol-3,4′-disulfate	Human [39]	antitumor, breast cancer [46]
Trans-resveratrol-glucuronide-sulfate	Mouse [40]	N
Dihydroresveratrol	Human [27]	antiproliferative, prostate cancer [47]
Dihydroresveratrol-glucuronide	Human [19]	N
Dihydroresveratrol-sulfate	Human [19] Mouse [40]	N
Dihydroresveratrol-glucuronide-sulfate	Mouse [40]	N
3,4′-dihydroxy-trans-stilbene	Human [27]	lower cholesterol level, cardiovascular [48]
Lunularin	Human [27]	lowering food intake, obesity [48]

**Table 2 molecules-25-04282-t002:** Effect of resveratrol on different targets in tumor microenvironment.

Tumor Microenvironment Targets	Resveratrol Effect	Mechanism of Action
Reactive oxygen species (ROS)	Inhibition and activation	Increases ROS generation [52] Scavenger of reactive oxygen species [58]
Tumor associated macrophages	Inhibition	Inhibition of M2 polarization of macrophage [66], inhibition of M2 macrophage activation [67], activates repolarization of macrophage from M2 to M1 phenotype [68]
Indoleamine 2,3-dioxygenase	Inhibition and activation	Inhibition of IDO expression and activity (50 mg/kg every 2 days for 3 weeks, animal study) [76] Increase in the activity of IDO (oral administration of 5 g resveratrol by 8 volunteers [77]
Vascular endothelial growth factor (VEGF)	Inhibition	Suppressing VEGF production [97] Inhibiting the production of HIF-1 [103]
Fibrosis	Inhibition	Inhibit activation, invasion, migration and glycolysis of cells involved in fibrogenesis process [113] Reducing several major fibrogenic mediators, such as a-SMA, type I collagen and fibronectin [114]
IL-6	Inhibition	Reduced IL-6-induced AR activity [127]

**Table 3 molecules-25-04282-t003:** Published human studies evaluating the effects of resveratrol on cancer cells as reported in the last 12 years.

Resveratrol Formulation	Dosage Administration	Cancer Type	Sample Size and Phase	Outcome of the Study	Ref
Resveratrol	20 or 80 mg/day for 14 days	Colon	N = 8 Phase 1	Reduction in the expression of a panel of Wnt target genes indicated inhibition of Wnt signaling in normal colonic mucosa	[131]
Grape Powder	80 or 120 g/day for 14 days				
Resveratrol	0.5 or 1.0 g for 8 days	Colon	N = 20	Quantification of Resveratrol and its metabolites found in colon tissue with a higher value obtained at right side of the colon. Tumor cell proliferation was reduced by 5% (Ki-67 immunostaining, *p* = 0.05).	[132]
Micronized resveratrol (SRT501)	5.0 g for 20 consecutive days in a 21-day cycle for a max of 12 cycles	Multiple myeloma	N = 24 Phase 2	Clinical trial conducted for assessment. SRT501 with or without bortezomib was given to multiple myeloma patients. Study was immediately terminated because of the appearance of serious adverse events and observed and minimal effectiveness in relapsed/refractory multiple myeloma patients. However, the previous outcome emphasized the risks of development and administration of the drug such populations.	[33]
Micronized resveratrol (SRT5001)	5 g for 14 days	Colorectal cancer patients with hepatic metastasis	N = 6	Resveratrol was detected on hepatic tissue and a 39% increase in the content of cleaved caspase-3 in malignant hepatic tissue.	[133]
Micronized resveratrol, (SRT501)	5.0 g for 14 days	Colorectal with hepatic metastases	N = 9 Phase 1	The detectable resveratrol in hepatic tissue that increased the cleavage of caspase-3 by 39% was well-tolerated with higher plasma resveratrol level (3.6-fold) in malignant hepatic tissue.	[134]
Pulverized muscadine grape skin Extract (MPX)	4000 mg/patient	Biochemically recurrent Prostate cancer patients	N = 14	MPX was safe but needs further investigation in d phase II trial with dose evaluation	[135]

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
