# Peer review of "Resveratrol and Tumor Microenvironment: Mechanistic Basis and Therapeutic Targets"

_molecules, 2020, doi:10.3390/molecules25184282_

Round 1

Reviewer 1 Report

In the proposed review submitted to Molecules-918164, the authors summarize the targets of resveratrol in the tumor microenvironment and the mechanisms involved in this targeting. The paper is interesting and brings insights regarding the physiological effects of resveratrol and its metabolites as a promising strategy to improve oncologic treatments, besides offering insights to new researches. However, this review could be enhanced, please find below my main concerns and suggestions.

Main concerns:

  • Line 59: In the topic 2, would be clear if the authors create a table summarizing the metabolites derived from resveratrol metabolism as well as the chemical variations naturally found through the plants and other substrates, the effects, possible local targeted and the diseases (mainly cancer) that were tested against the compounds. Exemplified in Resveratrol and Its Human Metabolites—Effects on Metabolic Health and Obesity https://doi.org/10.3390/nu11010143

  • Line 75-76: Authors cited this subject has been the focus of numerous in vivo and in vitro studies, but it was not mentioned in the manuscript. Please, cite the works and improve the discussion about it.

  • Line 617: I would strongly suggest reallocating the sentences written in Topic 8 (line 617) to the paragraph of lines 81-86.

  • Would better if change figure 3 to a table summarizing the subtopics presented. Describing the actions of resveratrol in each one, mainly focusing on possible molecules target. I believe it is going to be more informative to this journal’s readers.

  • Figure 2, although pretty, is not informative. Tumor cells do not form a mass and endothelial cells are not represented. Only ECM fibrillar proteins are represented. What is the taking home message in this figure?

A model to follow aiming to improve the information and schematization of figures (as suggested previously at comments 4 and 5) is to be based on the paper https://doi.org/10.3390/ijms20040925.

  • Line 595: Please, rewrite the sentence from lines 196-197 in topic 7, and increase the discussion on it.

  • I suggest writing putting together the subtopic 4.2 (line 198) and 4.3 (line 220), besides improving the discussion arguing over findings from experimental studies rather than reviews.

  • Line 323: Please, improve the discussion evidencing the resveratrol effects on fibrosis.

  • Line 606: Indicate the author in the phrase referencing the study number 139.

  • Line 593-594: Please, rewrite the sentence improving the argument of the industries' financial support to future studies.

  • The article counts with 52 references older than 10 years of publication and other newer comprising a large number of reviews. In this way, I suggest replacing some reviews to findings of recent experimental studies.

  • It is necessary a complete text review, mainly in punctuation markers and few English corrections, as well as potential plagiarism in some paragraphs along with the manuscript, as the example found it in lines 325-325 (highly similar to DOI: 1016/j.bbcan.2020.188356) and lines 504-509 (highly similar to DOI: 10.1016/j.bbadis.2014.11.004).

Author Response

Many Thank for your important comments. All your comments were considered and the manuscript was changed based on these comments. Please see attached our response to your comments.

Regards

Reviewer 2 Report

In the review entitled “Resveratrol and Tumor Microenvironment: 2 Mechanistic Basis and Therapeutic Targets”, the authors provide a view of the potential benefit of resveratrol on tumor inflammation and stromal cell signaling. They outline the structure, absorption and bioavailability of resveratrol which are underlying factors required to understand the activity and efficacy of resveratrol treatment for cancer pateints.  While this is an interesting and timely review, the clarity and organization of the writing makes it hard to follow what is known about resveratrol and what are the promising aspects of this treatment as an anticancer therapy.

Major concerns:

The section topics are well outlined but the text jumps around and is difficult to follow. It feels like quick summaries of the relevant studies, which shows the authors have done a careful literature review.  However, it would be nice to have a final paragraph in each section that gives the take home message for the reader (what is promising, key findings, and key studies that are still required). 

A main concept is that resveratrol is absorbed and metabolized rapidly and thus has very low bioavailability.  Even if bioavailability is low it must still have efficacy. It is know whether a different source (grape seed, micronized, etc) has altered bioavailability? Similarly, is there a dosing schedule that is more effective (3x/day vs once daily vs weekly)? 

Along these lines, discussion of the plasma level of resveratrol comes up several times throughout the review with relation to efficacy. I was still unable to parse out if there is a known plasma levels (concentration) that is required for efficacy.   

Section 5 outlines clinical trials with many severe side effects and lack of efficacy. Please clarify or reorganize this section to distinguish between resveratrol and other compounds. To follow, section 8 should reflect the safety outcome of resveratrol vs other compounds.

Minor:

Overall, the manuscript would benefit greatly from a language editor.

Figure 2 legend – Was this image created in Biorender?  If yes, it should be referenced.

Figure 3 legend is incomplete.

Author Response

Many thanks for your important comments. Your comments made a huge improvement in the manuscript.

All comments were considered and the manuscript was modified according to these comment. We hope this modified manuscript will meet your expectations.

Attached is our response to the comments.

Regards

Round 2

Reviewer 1 Report

Dear authors, the manuscript is readable and bring good information to this research field.

Finally, I suggest you check the format of table 1, unfortunately, it is pretty misaligned.

Reviewer 2 Report

The authors have much improved the review article and addressed all of my previous concerns.  The revised and additional table really help the reader synthesize the data presented in the manuscript. The manuscript is now acceptable for publication.